# A Comprehensive Experimental Investigation of Additives to Enhance Pool Boiling Heat Transfer of a Non-Azeotropic Mixture

**DOI:** 10.3390/e24111534

**Published:** 2022-10-26

**Authors:** Chen Xu, Zuoqin Qian, Jie Ren

**Affiliations:** School of Energy and Power Engineering, Wuhan University of Technology, Wuhan 430070, China

**Keywords:** non-azeotropic mixture, nanofluid, surfactant, critical mass fraction, mass transfer resistance

## Abstract

Adding nanoparticles or surfactants to pure working fluid is a common and effective method to improve the heat transfer performance of pool boiling. The objective of this research is to determine whether additives have the same efficient impact on heat transfer enhancement of the non-azeotropic mixture. In this paper, Ethylene Glycol/Deionized Water (EG/DW) was selected as the representing non-azeotropic mixture, and a comparative experiment was carried out between it and the pure working fluid. In addition, the effects of different concentrations of additives on the pool boiling heat transfer performance under different heat fluxes were experimentally studied, including TiO_2_ nanoparticles with different particle diameters, different kinds of surfactants, and mixtures of nanofluids and surfactants. The experimental results showed that the nanoparticles deteriorated the heat transfer of the EG/DW solution, while the surfactant enhanced the heat transfer of the solution when the concentration closed to a critical mass fraction (CMC). However, the improvement effect was unsteady with the increase in the heat flux density. The experimental results suggest that the mass transfer resistance of the non-azeotropic mixture is the most important factor in affecting heat transfer enhancement. Solutions with 20 nm TiO_2_ obtained a steady optimum heat transfer improvement by adding surfactants.

## 1. Introduction

As a heat transfer process with a phase change, pool boiling has been widely used in many industrial fields [1]. With the development of the industry, the effective method of passive enhanced boiling heat transfer to achieve a higher heat transfer coefficient with a small area has gradually become a research hot spot [2]. In addition to changing the heat exchange equipment to enhance the heat transfer, the traditional working fluids can not meet the heat transfer demands today [3]. Therefore, more efficient late-model working fluids need to be explored.

Choi [4] brought up the idea of nanofluids for the first time. After that, boiling heat transfer characteristics and applications of various nanofluids have been studied in lots of research. Many systems consider nanofluids as preferred working mediums for their merits. Rahimi et al. [5] brought the novel idea of applying nanofluid into redox flow batteries, which led the way to the development of high-efficient and low-cost batteries. Colangelo et al. [6] suggested that using nanofluids as the heat transfer fluid of electronic devices could not only optimize the dimensions of the electronic devices but also obtain higher energy efficiency at the same time. The majority of the literature thought a small number of nanoparticles can greatly improve the boiling heat transfer characteristics of the solution.

At the same time, a certain number of researchers thought some features of nanofluids after boiling had a significant effect on the boiling characteristics. The size and concentration of the nanofluid will influence the increase, decrease, or in some cases, no effect on pool boiling heat transfer. Hu et al. [7] experimentally investigated the boiling heat transfer characteristics of the mixture of different diameters and concentrations of SiO_2_ nanoparticles with a 60% ethylene glycol (EG) aqueous solution. The results of all the experiments showed that nanofluids have a higher heat transfer coefficient (HTC) than the EG aqueous solution, even though the HTC deteriorated for nanoparticle volume fractions above 0.75%. Moreover, the HTC increased with the decrease of the nanoparticle diameter. Peng et al. [8] prepared three different average diameters of Cu nanoparticles to be mixed with an R113/oil mixture as a test fluid. The HTC of the experiments presented an ascend result not only with the decrease of the nanoparticle diameter but also with the increase of the nanoparticle concentration.

However, contrary results were reported by Trisaksri et al. [9], who studied the nucleate pool boiling heat transfer characteristics of TiO_2_/HCFC 141b nanofluid at different concentrations. The HTC deteriorated with the increase of the nanoparticle concentration, and the degree of deterioration augmented with the increase of the heat flux. Naphon et al. [10] experimentally studied the HTC of the nanoparticle TiO_2_, suspended in ethyl alcohol mixed with refrigerant R141b as the base fluid. It was found that the HTC was inversely proportional to the concentration of the nanoparticles, especially at high heat fluxes.

Azimi et al. [11] considered that nanoparticles’ deposition would cause different effects due to corresponding thermal conductivities. The effective thermal conductivity of the nanofluid was enhanced with the increasing volume concentration so that the effects of the deposition layer could be neutralized. According to the investigation of bubble growth behavior in nanofluids with different concentrations by Morad et al. [12], the bubble behavior had a significant impact on the thermal efficiency of the solution, while the surface tension and viscosity of the nanofluid solution played a crucial role in the bubble radius and thermal diffusivity. Abu-Nab et al. [13] also emphasized the importance of the influence of surface tension on the dynamics of nanofluid bubbles. In order to stabilize the enhanced heat transfer ability and reduce the surface tension of nanofluids, many researchers choose to add a small ratio of different surfactants into the nanofluids.

Etedali et al. [14] experimentally studied the performance of the heat transfer of a SiO_2_/DW nanofluid with added surfactants (Ps20, CTAB, and SLS) on a copper surface, and observed better stabilization of nanofluids. It was found that nanofluids with surfactants have a higher HTC because the lower surface tension enhances the bubble separation speed. Jung et al. [15] investigated the effects of adding nitric acid into TiO_2_/water nanofluids on CHF in pool boiling. The experimental results showed that the additive induced a more even dispersion of nanoparticles in the nanofluids.

Tang et al. [16] measured the boiling heat transfer characteristics of 0.001 vol.%, 0.01 vol.%, and 0.1 vol.% δ-Al_2_O_3_/R141b nanofluids with and without surfactant SDS. They found that nanofluid with SDS had increasing boiling HTC and decreasing boiling deterioration caused by the deposition of a high-volume concentration nanofluid. The pool boiling heat transfer of a silica DI water nanofluid with three various surfactants (SDS, CTAB, and PS20) was reported by Tian et al. [17]. According to the results of the experiments, nanofluids with an anionic surfactant (SDS) had the maximum increased HTC and boiling surface roughness.

Peng et al. [18] observed that surfactant additives enhanced the nucleate pool boiling heat transfer of nanofluids but deteriorate the nucleate pool boiling heat transfer at high concentrations and of each type of surfactant presenting the maximum value at an optimal concentration. Khooshechin et al. [19] used SDS as a surfactant additive to overcome the instability of nanoparticles by increasing the boiling temperature and time. However, they found that boiling the HTC only enhanced by surfactants in low heat fluxes and surfactant particles may cause deposition on the heater surface.

From a practical point of view, good stability of the nanofluid is essential to have effective thermal-flow systems. As described above, although surfactant additives can enhance the stability and the pool boiling heat transfer performance of nanofluids in some conditions, controversial results still could not be avoided, especially when the basic fluid of the nanofluid was a mixture.

There have been numerous studies to investigate heat transfer enhancement in a pure working medium, including adding nanoparticles and a surfactant into the pure basic fluid. However, the application of a pure working fluid has limitations, especially pure refrigerants which normally have a high Global Warming Potential (GWP), which will cause great harm to environmental protection. Recently, researchers have shown an increased interest in binary mixtures. Li et al. [20] considered that, through a reasonable adjustment in the composition, binary mixtures would combine merits such as safety, environmental protection, and high-efficiency performances. 

The experimental results of Shah et al. [21] showed that the heat transfer coefficient of the mixed working medium decreased. In fact, the temperature slip caused by the boiling of different components and the mass transfer resistance between the components [22] influenced the potential application of the mixed working fluids. 

Therefore, an important issue in the application of a binary mixture in pool boiling is that the deterioration of the HTC caused by boiling urgently needs to be improved. The purpose of the present research is to figure out the best way to enhance the pool boiling performance of a non-azeotropic binary mixture through the carrying out of comprehensive experimental investigations about the influence of the EG concentration, surfactant type, and nanoparticle size.

## 2. Experimental Setup

### 2.1. Experimental Apparatus and Procedure

The experimental apparatus consists of four main parts, including a power control system, solution boiling system, data acquisition system, and heating copper rod apparatus. The schematic diagram of the boiling heat transfer apparatus is shown in Figure 1.

The power control system includes an AC power supply, AC automatic voltage regulator, AC voltage regulator, and intelligent power meter. The input heat load controlled by the voltage regulator passes through the regulator, and the input value is clearly displayed on the power meter and then loaded on the copper column heating device. After heating, the data acquisition system collects the relevant experimental data, including the temperature sensor that displays the real-time temperature through the digital display controller, the high-speed industrial camera that records the dynamic changes of the bubbles during boiling, and the computer that collects and records the experimental data.

The working fluid boiling system is mainly composed of a square container made of high borosilicate glass, with an external dimension of 10 × 10 × 10 cm^3^. The glass reflection cover above the container is connected to a condensation pipe, a preheating heating rod, and a temperature sensor. Among them, the preheating heating rod is a special-made high-temperature and corrosion-resistant heating rod, and the condensing pipe is connected to an external circulating water pump for cooling. Observation ports are reserved on both sides of the container, and the rest are wrapped by high-temperature insulation materials. A 2 × 2 cm^2^ square hole under the container is reserved for the heated surface on the top of the heating column processed from red copper. Seven cylindrical holes for placing heating rods are reserved under the copper column, and the specific dimensions are shown in Figure 2. In order to ensure the accuracy of the experiment, the copper column heating device is also wrapped with thermal insulation materials, and Teflon plates are installed at the connection with the boiling vessel to increase the tightness and thermal insulation.

First of all, before the installation of the equipment, the heating surface was polished with ultra-fine sandpaper #2000 and polished with a polishing machine. We used high-temperature heat transfer oil and a high-temperature mercury thermometer to correct the temperature of the temperature sensor. Secondly, the heated copper surface was scrubbed with a copper detergent before each experiment, and the entire solution cavity was cleaned with deionized water. Then, we injected the working fluid and marked the height of the liquid level surface to ensure the repeatability of the experiment.

At the beginning of the experiment, we used the preheating heating rod to preheat the working medium in the container to the saturation temperature to discharge the insoluble gas in the working medium. At the same time, the heating rod works at low power at first. When the value of the temperature sensor changes less than 1 °C within 5 min, we gradually increased the heating load.

### 2.2. Experimental Data Reduction and Uncertainty Analysis

Since the heating and heat conduction sections are insulated with insulating materials, the heat flux, q, could be calculated as a stabilized one-dimensional heat conduction by the Fourier law:(1)$$q=\lambda \left[\left(T_{3}-T_{1}\right)/\left(x_{3}-x_{1}\right)+\left(T_{2}-T_{1}\right)/\left(x_{2}-x_{1}\right)+\left(T_{3}-T_{2}\right)/\left(x_{3}-x_{2}\right)\right]/3$$
where λ is the thermal conductivity of the heating rod, T_i_ (i = 1, 2, 3) is the temperature of the measuring point, and x_i_ is the distance between the measuring point and the top heated surface.

Then, the boiling heat transfer coefficient, h, could be obtained from the following formula:(2)$$h=q/\left(T_{\mathrm{ave}}-T_{\mathrm{sat}}\right)$$
where T_sat_ is the saturation temperature of the solution, which is measured by a thermocouple placed 2 cm above the heating surface, and T_ave_ is the average temperature of three measuring points calculated from Equation (3): (3)$$T_{\mathrm{ave}}=\sum_{i=1}^{2}[T_{i}\delta -x_{i}\left(T_{i+1}-T_{i}\right)]/2\delta$$
where δ is the distance between the two measuring points.

The uncertainty in this study was mainly caused by thermocouples with an accuracy of 0.1 K. This negligible change in the data acquisition system used was considered as no uncertainty. The uncertainty of the nucleate pool boiling heat transfer coefficient was estimated by Kline and McClintock [23]. The maximum uncertainties of the heat flux, heating surface temperature, and heat transfer coefficient in this research are within 8.6%. 

In order to verify the reliability of the experimental system, the widely studied deionized water was selected to carry out the pool boiling experiment, and the experimental results were compared with the Rohsenow curve used in other research, and the errors were within ±10%. It can be seen from Figure 3 that the experimental results are very close to the experimental results of Norouzipour [24].

## 3. Characteristics, Selection, and Preparation of Working Fluids

### 3.1. Non-Azeotropic Mixtures

Through a lot of research by scholars, it is known that surface tension, shear viscosity, and thermal conductivity are the main factors affecting the boiling characteristics of working fluids. For non-azeotropic mixtures, different components have their respective saturation boiling temperatures, which will cause a temperature slip and induce additional mass transfer resistance. This is also the reason why the boiling heat transfer performance of the non-azeotropic mixed solution is generally lower than that of pure working fluids. It can be seen from Figure 4 that the surface tension of EG/DW mixed solutions decreases with the increase of the volume fraction of the ethylene glycol. To well known, the reduction of the surface tension was supposed to reduce the bubble diameter generated during boiling, so as to strengthen the boiling heat transfer performance.

However, the boiling experiment results in Figure 5 show that the heat transfer performance decreases with the increase of the ethylene glycol ratio. Therefore, enhancing the boiling heat transfer performance of non-azeotropic mixed solutions cannot be simply obtained by reducing the surface tension.

Meanwhile, with the increase of the ethylene glycol concentration, the saturation temperature of the solution will also change, which will delay the initial boiling temperature point.

Figure 6 shows the bubble growth behavior during boiling. The diameter of the bubble decreased with the concentration increase since the surface tension decreased. At the same time, the bubble departure frequency also decent caused by the growing mass transfer resistance between components.

Therefore, in this paper, a 30% EG/DW solution was selected as the base fluid to study pool boiling-enhanced heat transfer since it can effectively reflect the enhanced effect of subsequent experiments on the boiling performance of the EG/DW solution, while not affecting the control experiment due to the excessive mass transfer resistance.

### 3.2. Properties of the Surfactant

In order to study the influence of different kinds of surfactants on the boiling performance of the base fluid, three different kinds of typical surfactants that are widely used were selected for the experiments: the cationic surfactant, CTAB, anionic surfactant, SDS, and a non-ionic surfactant, Triton X-114. Among them, CTAB and SDS are dry white powders and Triton X-114 is a viscous liquid, and the purity was more than 99.9%. We used an electronic scale with an accuracy of 0.0001 g to weigh the surfactants of different concentrations and added them to the base solution. Then, an electromagnetic mixer was used to stir for 6 h to make the solution mix evenly and stably.

Based on the literature of other scholars [25,26], the optimal concentration ranges of a surfactant aqueous solution can be found, and three surfactant concentrations were determined as follows: CTAB (200~600 ppm), SDS (1500~2500 ppm), and Triton X-114 (200~400 ppm).

### 3.3. Preparation of Nanofluids

In this paper, the average size of the TiO_2_ nanoparticles was 20 nm and 50 nm. After an ultrasonic bath, a two-step method was adopted by dispersing different weight concentrations of TiO_2_ into the base fluid with an electromagnetic mixer, stirring for 6 h. The preparation of the nanofluids with two particle diameters at different volume fractions was 0.001%, 0.01%, and 0.1%. 

In addition, since nanoparticles are unstable and easy to agglomerate, the experiments with a mixture solution of the nano-based fluid (the 0.001% EG/DW nanofluids) and a surfactant were carried out to explore the enhancement of the non-azeotropic mixtures.

## 4. Results and Discussions

Thirty-three groups of experiments were carried out under atmospheric pressure. Besides the base solution (EG/DW = 30:70), we also added TiO_2_ nanoparticles with particle diameters of 20 nm and 50 nm in different concentrations, three representative surfactants in different concentrations, and 0.001% nanofluids in different particle sizes with various surfactants. 

### 4.1. Effect of Nanoparticles on Nucleate Pool Boiling Heat Transfer

Figure 7 presents the pool boiling heat transfer coefficients versus the heat flux for the base fluids with 20 nm and 50 nm TiO_2_ nanoparticles in different concentrations. It can be seen from the figure that the initial boiling point of all the EG/DW solutions added with the nanoparticles is delayed, and the boiling heat transfer performance decreases in various degrees. Besides, the degree of decline increases with the increase of the superheat temperature.

In addition, it can be found that when the particle diameter is 20 nm, the effect of increasing the concentration of the nanoparticles on the boiling heat transfer performance of the solution increases with the increase of the particle concentration. However, when the particle diameter is 50 nm, the effect of the particle concentration on the heat transfer performance of the solution does not have a certain regularity, but when the superheat is high, the heat transfer performance of the 0.1% nanofluid is the worst.

The main reason for the enhanced heat transfer of nanofluids is that nanoparticles can be adsorbed on the gas–liquid interface to increase the stability of bubble formation. At the same time, the number of bubble nucleation points is also increased in the deposition layer on the heating surface caused by the nanoparticles [27]. Therefore, the possible reason for the heat transfer degradation is that the large mass transfer resistance of the EG/DW solution during boiling makes the nanoparticles tend to agglomerate rather than adsorb to the gas–liquid interface. Therefore, the addition of the nanoparticles hinders the formation of the bubbles during boiling, resulting in the backwardness of the initial boiling point. In addition, the resulting increase in dynamic viscosity also inhibits the deposition of the nanoparticles on the heating surface, especially when the particle diameter is smaller, and the deposition is lesser. This also explains why the overall heat transfer deterioration effect of 20 nm nanofluids is stronger than that of 50 nm nanofluids.

### 4.2. Effect of Surfactant on Nucleate Pool Boiling Heat Transfer

The change in the pool boiling heat transfer enhancement with a concentration for the three surfactants is graphed in Figure 8. In general, EG/DW with the addition of the surfactants enhances the heat transfer performance of the solution, despite the kind of surfactant. 

It can be found that the heat transfer effect of the three surfactants is most improved when the concentration nears the CMC. At a low heat flux, the solution with 400 ppm CTAB and 2000 ppm SDS has the best boiling heat transfer performance, and the heat transfer effect is far better than that with TritonX-114. At the same time, all the concentrations of CTAB and SDS show an early onset boiling point compared with the base fluid. As to TritonX-114, it only shows the early onset boiling point when it nears the CMC concentration. However, as the heat flux climbs, the heat transfer enhancement of TritonX-114 gradually becomes the best. When the heat flux exceeds 100 kW/m^2^, the heat transfer performance of the solutions with CTAB and SDS in low concentrations gradually deteriorates. Significantly, when the heat flux exceeds 270 kW/m^2^, all the concentrations of the solutions with CTAB will no longer improve the heat transfer performance.

It is well known that the addition of surfactants will reduce the surface tension of the solution, reduce the bubble diameter, and increase the generation frequency of the bubbles, and hence enhance the boiling heat transfer. Especially when the concentration is close to the CMC, the performance of the heat transfer has an optimal enhancement.

In addition, the surfactants also have a good adsorbing capacity at the gas–liquid interface [28], which has a significant impact on the boiling heat transfer of the non-azeotropic mixtures. At a low heat flux density, the surface tension is the main factor affecting the heat transfer, so the CTAB and SDS solutions in nearly CMC concentrations have an obvious effect on heat transfer enhancement. With the continuous increase of the heat flux density, the bubble density and frequency also rise, and the effect between the bubble and liquid becomes more obvious, which grows the mass transfer resistance between the components [29]. At this time, the interfacial adsorption of the surfactant becomes the main factor affecting the heat transfer performance. Therefore, the boiling heat transfer enhancement effect of the non-ionic surfactant TritonX-114 solution is gradually enhanced.

### 4.3. Effect of Adding Surfactant into Nanofluids on Nucleate Pool Boiling Heat Transfer

The heat transfer curves of the 0.001% 20 nm and 50 nm TiO_2_ nanofluids with added CTAB in different concentrations are illustrated in Figure 9. It can be seen that the two solutions with different nanoparticle diameters show completely different heat transfer effects. For the 20 nm nanofluids, the addition of 200 ppm CTAB significantly improved the onset boiling point and boiling heat transfer capacity of the solution. With the increase of the CTAB concentration, the heat transfer enhancement decreases gradually. Even so, the heat transfer capacity of the solution is improved compared with the base liquid. For the 50 nm nanofluids, the boiling heat transfer capacity increases with the increase of the CTAB concentration. However, even if the heat transfer capacity of the mixed solution is improved, the addition of CTAB does not offset the heat transfer deterioration caused by the addition of the nanoparticles, and the heat transfer capacity of the 50 nm nanofluids are all lower than that of the EG/DW solution.

Figure 10 depicts the boiling heat transfer curve of the nanofluids with SDS added. The nanofluids with two particle diameters also show two different heat transfer enhancement performances. For the 20 nm nanofluids, adding SDS can effectively improve the heat transfer capacity of the solution. Among them, the solution that added SDS with a concentration close to the CMC obtained the optimum boiling heat transfer enhancement. However, for the 50 nm nanofluids, the heat transfer performance of the solution can achieve a slight enhancement only when there is a low concentration of SDS added. In addition, the continuous increase of the concentration of SDS will only make the boiling heat transfer constantly decline.

The heat transfer curve of the nanofluid added with TritonX-114 is shown in Figure 11. Compared with the first two kinds of surfactants, TritonX-114 has a significantly different effect on the boiling heat transfer of the solution. No matter what the diameter of the nanoparticles is, the addition of TritonX-114 has no enhancement on the boiling heat transfer performance of the nanofluids, and the heat transfer deterioration enlarges with the increase of the concentration.

From the above solution boiling curves, it can be found that the nanofluid with CTAB or SDS can not only achieve better boiling heat transfer performance but also tend to be more stable than the base liquid with the surfactants. The possible reason is that the adsorption of the surfactants is employed to inhibit the agglomeration of the nanoparticles instead to disturb the gas–liquid interface. Hence, the formation and separation of the bubbles tend to be more stable. The reduction rate of the surface tension has become the main factor affecting the heat transfer performance. However, since the surfactants also affect the contact angle of the nanoparticles and the deposition of the particles on the heated surface, the reduction of the surface tension is not the only reason that affects the improvement of the pool boiling heat transfer performance. This is also why the nanofluid added with CTAB does not have the highest heat transfer performance when the concentration is close to the CMC. Therefore, no matter what the concentration of TritonX-114 is, although it can improve the heat transfer deterioration of the nanofluids to a certain extent, the heat transfer performance of the non-azeotropic mixed base fluids cannot be improved because of the slow reduction rate of the surface tension.

In addition, no matter what kind of surfactant is added, the nucleate boiling heat transfer performance of the 20 nm nanofluids is much better than that of the 50 nm nanofluids. The reason of changing effects on the heat transfer enhancement largely results from the fact that more bubble nucleation points on the heated surface formed due to the smaller nanoparticle size. What is more, the addition of the surfactants amplifies this effect, which induces the heat transfer performance to increase with the decrease of the diameter of the nanoparticles. The change in the surface roughness of the heating surface after boiling in the various solutions can be seen in Figure 12.

## 5. Conclusions

In this study, EG/DW was selected as the representative of a non-azeotropic mixture as the basic working fluid, and its pool boiling experiment was carried out. The study offers some important insights into the pool heat transfer performance enhancement of a non-azeotropic mixture, which has a significant potential application in small electronic devices. However, additional studies to understand more completely the key tenets of a mixture with additives are required, including the influence of the changing physical properties of fluids in various concentrations. 

The experimental results have confirmed the heat transfer degradation phenomena of the non-azeotropic mixture compared with the pure refrigerant during pool boiling. At the same time, the pool boiling characteristics of the mixed solution with nanoparticles and surfactants were experimentally studied, and the major conclusions after the observation and analysis of the experimental results are listed below.

The increase of the ethylene glycol volume fraction in the EG/DW mixture can reduce the surface tension of the solution. However, the increase of the mass transfer resistance between the components and between the gas and liquid interface during boiling will still cause the heat transfer degradation of the solution;Instead of the pure working fluid with nanoparticles having an excellent effect on the pool boiling heat transfer performance, adding nanoparticles to the EG/DW solution deteriorated the heat transfer performance. The factor that contributes to this situation is that nanoparticles tend to cluster when boiling since the mass transfer resistance of non-azeotropic mixtures has a negative impact on the Brownian motion of the nanoparticles. Although increasing the diameter of the nanoparticles can reduce the deterioration to some degree, continuously increasing the concentration of the nanoparticles will cause more serious degradation of the heat transfer.Adding the surfactant enhanced the pool boiling heat transfer performance of the EG/DW non-azeotropic mixture. However, the heat transfer performance of the solution with the surfactant was not only related to the type and concentration of the surfactant but was also related to the heat flux. The heat transfer performance of the solution with 400 ppm CTAB and 2000 ppm SDS added to it had the most obvious enhancement at a low heat flux, while the optimal heat transfer enhancement belonged to the solution with 300 ppm TritonX-114 when the heat flux density exceeded 200 kW/m^2^. It is not simple to give the reason for this complicated phenomenon. Of course, the surface tension reduction is not the sole reason for affecting the heat transfer enhancement. The change in the heat transfer enhancement at a high heat flux largely resulted from the adsorption capacity of the surfactants’ effect on the interfacial mass transfer resistance and bubble dynamics.Compared with adding surfactant solely in the EG/DW solution, adding surfactants in the mixed solution with 20 nm TiO_2_ nanoparticles obtained a greater boiling heat transfer improvement. However, this improvement effect was limited to the addition of the cationic surfactant CTAB and anionic surfactant SDS. The main explanation accounting for the phenomenon is that the surfactant promoted the Brownian motion of the low-concentration nanoparticles of a small particle diameter since the mass transfer resistance of the mixed solution was suppressed. Therefore, the main factor affecting the heat transfer performance goes back to the reduction rate of the surface tension. At the same time, the deposition layer of the small-sized nanoparticles on the heated surface increased the nucleation points of the bubbles and further enhanced the heat transfer performance.

## Figures and Tables

**Figure 1 entropy-24-01534-f001:**
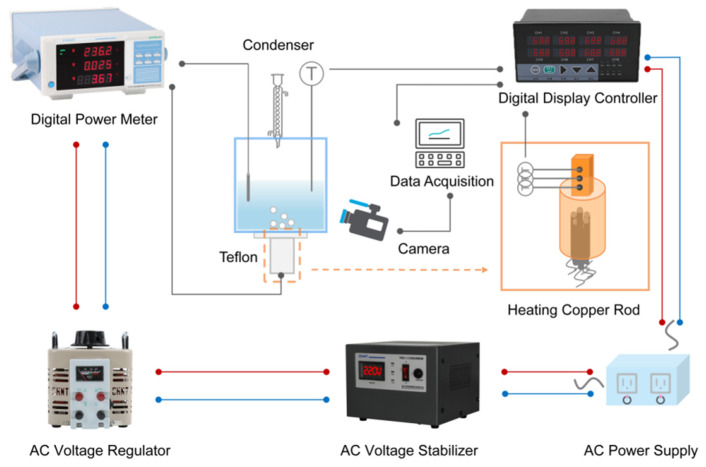
A scheme of the experimental procedure.

**Figure 2 entropy-24-01534-f002:**
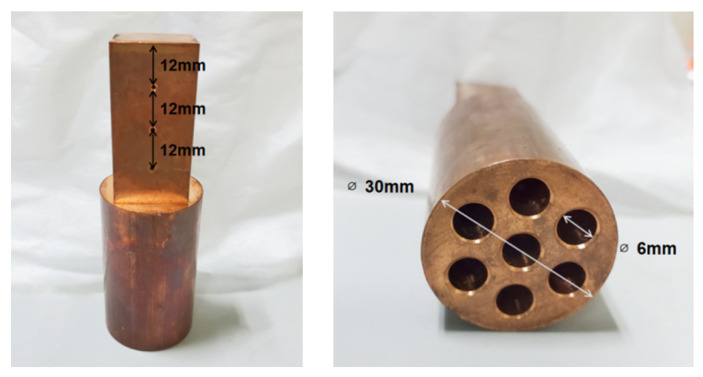
Schematic diagram of the copper heating column.

**Figure 3 entropy-24-01534-f003:**
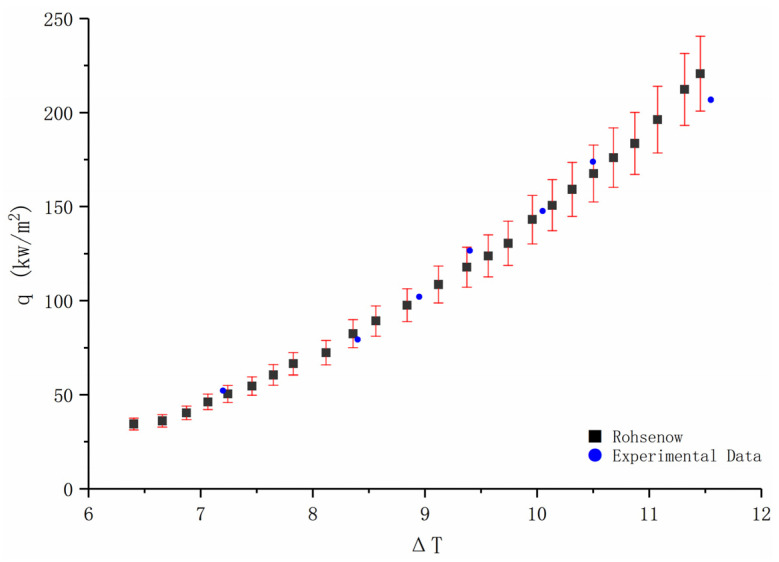
Validation of the experimental results.

**Figure 4 entropy-24-01534-f004:**
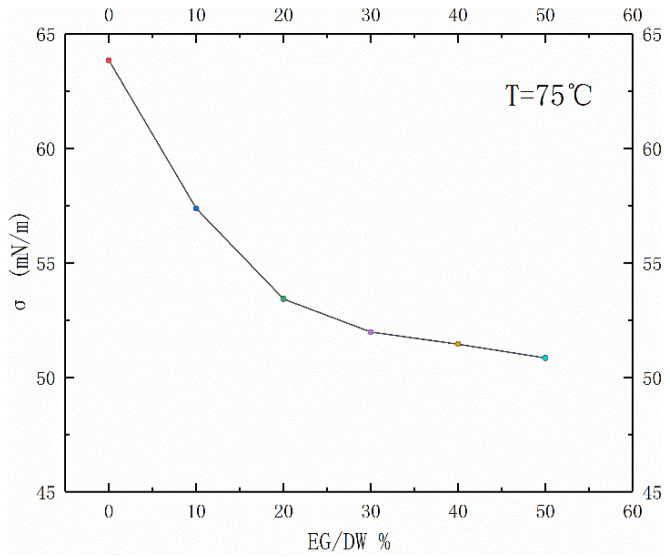
Concentration effect of the EG/DW solution on surface tension.

**Figure 5 entropy-24-01534-f005:**
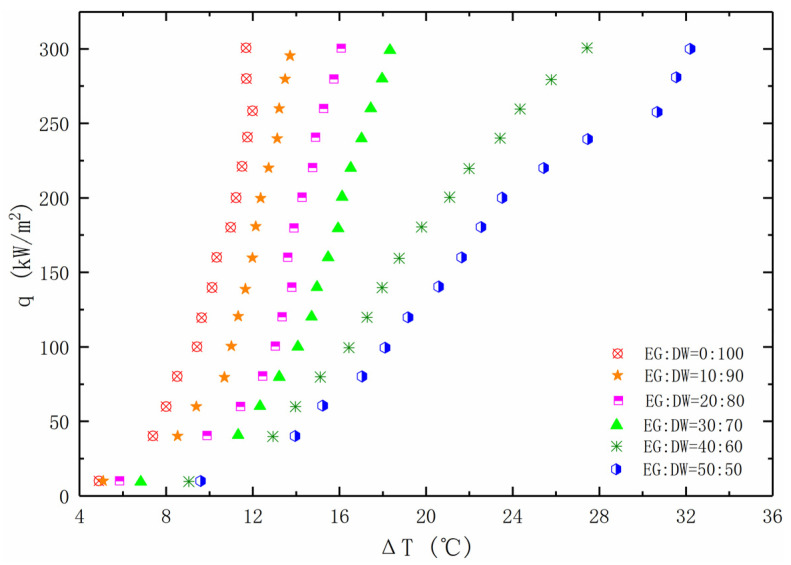
Concentration effect of the EG/DW solution on the boiling curve, plotted as the heat flux against the wall super heat temperature.

**Figure 6 entropy-24-01534-f006:**
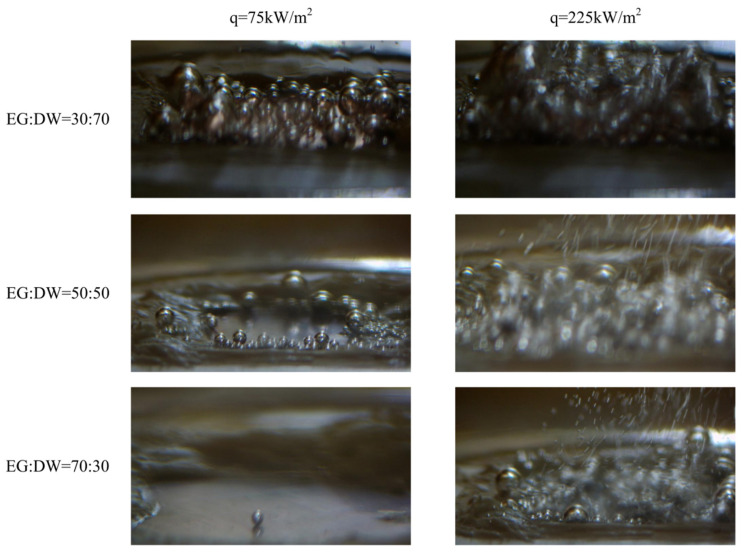
Bubble growth of various concentrations of the EG/DW solution under different heat fluxes during boiling.

**Figure 7 entropy-24-01534-f007:**
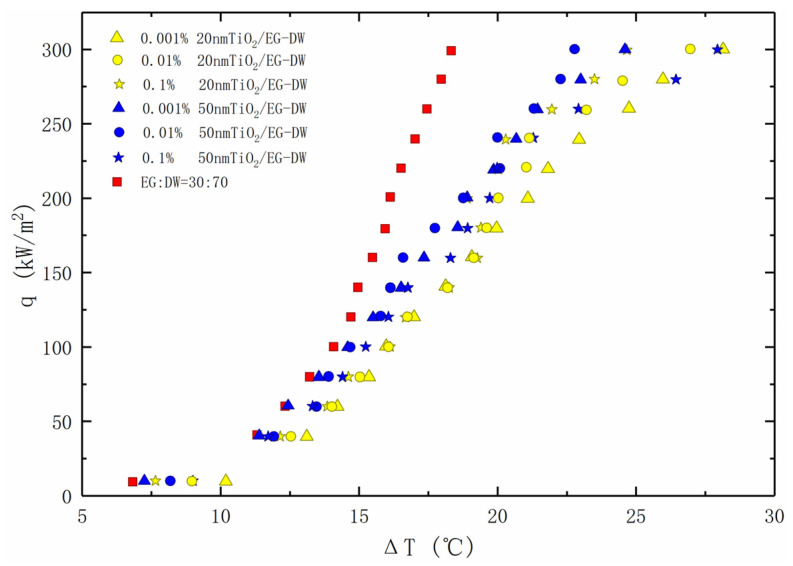
Nucleate pool boiling heat transfer data for nanofluids, plotted as the heat flux against the wall super heat temperature.

**Figure 8 entropy-24-01534-f008:**
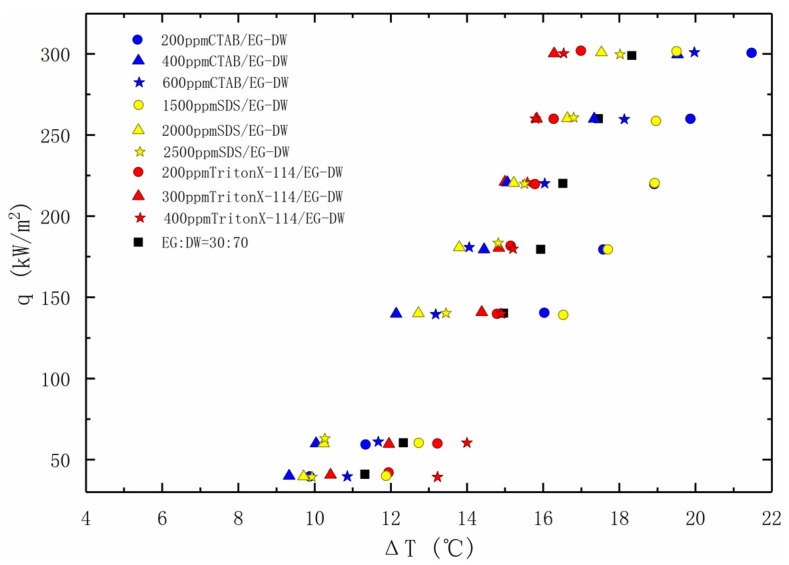
Nucleate pool boiling heat transfer data for solutions with various surfactants, plotted as the heat flux against the wall super heat temperature.

**Figure 9 entropy-24-01534-f009:**
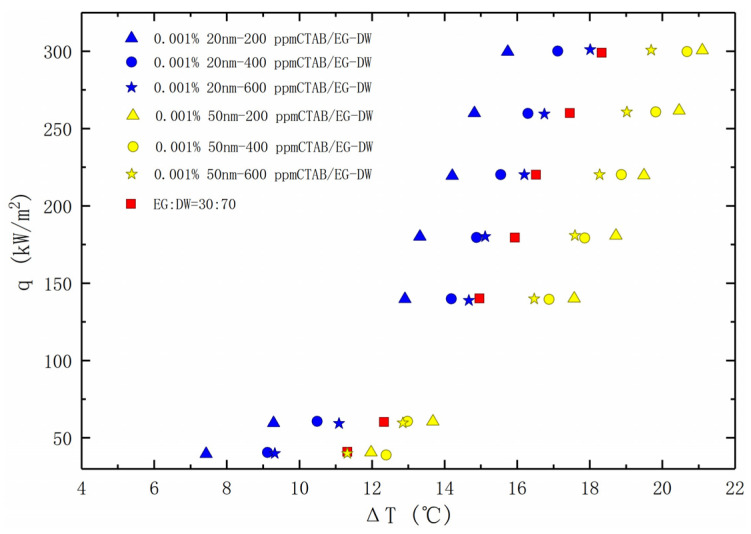
Nucleate pool boiling heat transfer data for nanofluids with CTAB, plotted as the heat flux against the wall super heat temperature.

**Figure 10 entropy-24-01534-f010:**
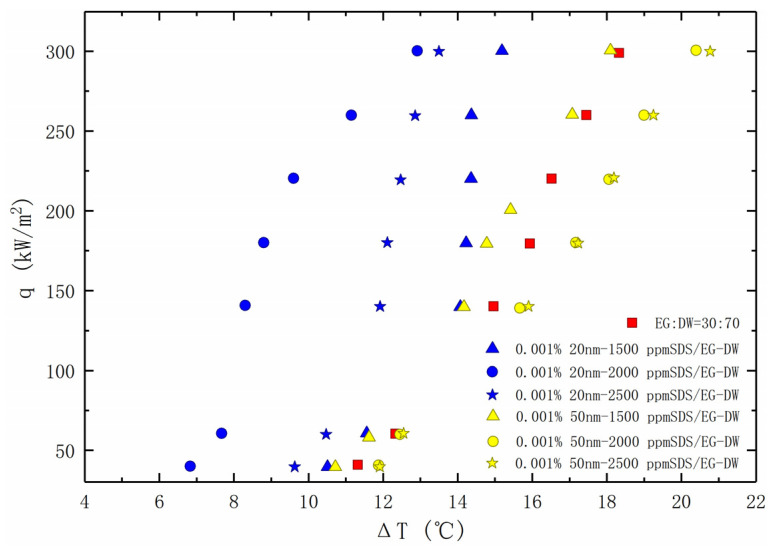
Nucleate pool boiling heat transfer data for nanofluids with SDS, plotted as the heat flux against the wall super heat temperature.

**Figure 11 entropy-24-01534-f011:**
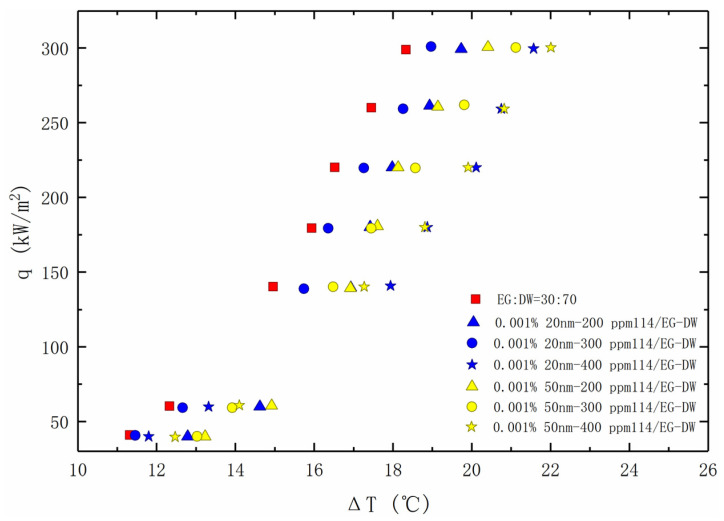
Nucleate pool boiling heat transfer data for nanofluids with TritonX-114, plotted as the heat flux against the wall super heat temperature.

**Figure 12 entropy-24-01534-f012:**
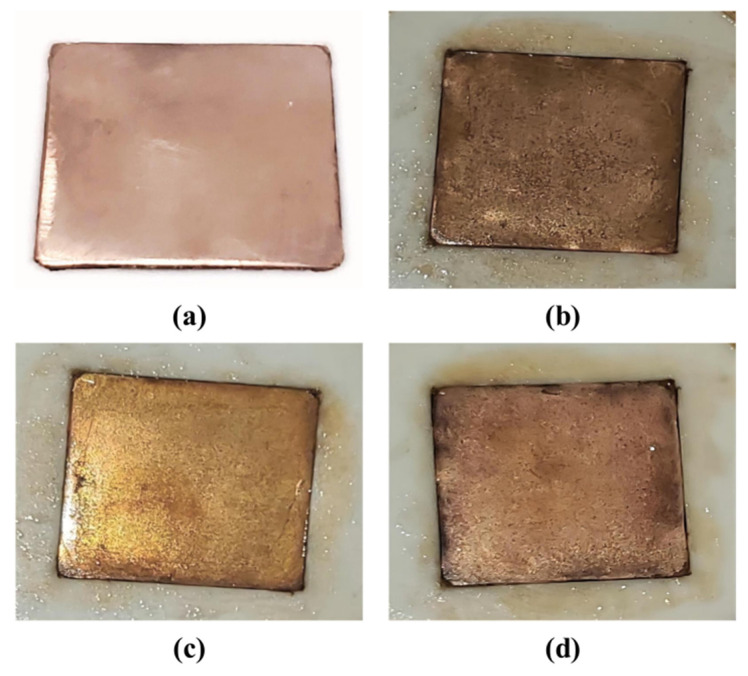
(**a**) Plain heated surface before boiling versus the heated surfaces after nucleate pool boiling in 0.001% EG/DW/TiO_2_ nanofluids with various surfactants: (**b**) CTAB; (**c**) SDS; (**d**) TritonX-114.

## Data Availability

Not applicable.

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
