# Peer review of "A Comprehensive Experimental Investigation of Additives to Enhance Pool Boiling Heat Transfer of a Non-Azeotropic Mixture"

_entropy, 2022, doi:10.3390/e24111534_

Round 1

Reviewer 1 Report

Comments to the Editor and Author

Manuscript Number: entropy-1887117

Comments on “Experimental investigation of additives to enhance pool boiling heat transfer of non-azeotropic mixture”. By Chen Xu, Zuoqin Qian, Jie Ren.

In the present paper, the authors determined whether additives have an efficient impact on heat transfer enhancement of non-azeotropic mixture. They selected Ethylene Glycol/De-ionized Water (EG-DW) fluids. They also studied the effects of different concentrations of additives on the pool boiling heat transfer performance under different heat flux, including TiO2 nanoparticles with different particle diameters, different kinds of surfactants, and mixtures of nanofluids and surfactants. The experimental results showed that the nanoparticles deteriorated the heat transfer of EG-DW solution.

To some sense, the study certainly addresses a relevantly new study, and it also delivers some results that might be of interest to the readership of the Entropy journal. Unfortunately, the presentation of the material is not quite sufficient and relevant preceding work is not given credit. The authors also need to specifically describe the exact contribution of their work compared to published works. A minor revision is needed to improve the manuscript. Comments that should be addressed thoroughly are:

Comments #01: The introduction section lists a lot of other people's research work, but it does not impress on readers the importance of this work, what is not solved in this field and why is it important. The authors need to discuss the previous work instead of only mentioning that “Many systems consider nanofluids as preferred working mediums for their merits, including batteries [5], cooling of electronics [6], nuclear reactors [7], and more”.

Comments #02: Some of the results and conclusions of this paper are quite basic. I recommend expanding introduction, results, and discussion sections. The aim should be to: 1) give a broader view of the literature on the topic and the current state-of-the-art; 2) clarify and discuss the novelty and the significance of the results obtained here, and compare them with those available in the literature, also including discussions on potential applications; 3) complete the manuscript with some additional, less basic results.

Comments #03: The authors need to provide a more convincing motivation for studying this problem. The authors can add the following reference of thermophysical properties of nanofluids using theoretical methods “https://doi.org/10.1016/j.csite.2021.101527, https://doi.org/10.1088/1402-4896/abdb5a, 10.1088/1402-4896/ac8bb2

Comments #05: Authors should not only state what is shown, but it should be made clear to readers why the figure has been included and what is of interest.

Comments #06: Should explain how accurate the results as compared to the earlier results (with other methods).

Comments #07: The validity and reliability of the present model should be verified.

Comments #08: In figures 4-11 it is not clear what kind of waves these figures show.

Can the authors explain what are they showing in these figures?

Comments #09: The motivation of this work is well presented; however, the physical implications of the results, in my opinion, are not sufficiently emphasized.

In conclusion, this paper is theoretically detailed, has some strength, and well written. However, it has shortcomings, as listed above, that should be covered before publication.

Author Response

Dear reviewer:

Thank you for your decision and constructive comments on my manuscript. We have carefully considered the suggestion of the Reviewer and made some changes. We have tried our best to improve and made some changes to the manuscript.

The yellow part has been revised according to your comments. Revision notes, point-to-point, are given as follows:

1. Thank you for your kind suggestions. We rewrote the Introduction section according to Coments#01, 02, 03, and 09. The detailed changes are the following:

  • We adjusted the overall logic of this part to make the introduction clearer, and made a more detailed summary of the previous researchers' research content so that readers can understand it more clearly;
  • We highlighted the research focus of the paper, namely, the application prospect of mixed working fluids, and the importance of how to improve the boiling heat transfer performance of mixed working fluids;
  • We have added some updated and more convincing references, including the two papers you recommended, to better elaborate the theoretical research;

2. Comments#06 and 07:

We appreciate your opinion on this section. The validation of the experimental setup had been compared in Figure 3 with other researcher's study. And we stated the errors more specifically.

3. Comments#02:

We appreciate your opinion on this section. The results and discussions had been revised. The details are underlined in yellow.

4. Comments#05 and 08:

We apologize if our original figures did not show the experimental results clearly. We supplemented the figures.

We would like to express our great appreciation to you for your comments on our paper. Looking forward to hearing from you soon.

Thank you and best regards!

Yours sincerely,

Chen Xu

Reviewer 2 Report

Several studies have shown that nanofluids improve critical heat flux under pool boiling conditions due to deposits of the nanoparticle on the heater surface. The authors propose to contribute with more information regarding the impact on heat transfer enhancement of nanofluids with TiO2 and/or surfactants in a non-azeotropic mixture of Ethylene Glycol/De-ionized Water. There is little experimental data for the case of these nanofluids mixed with surfactants in comparison to nanofluids or surfactant solution and more extensive research in this field is required.

Adding nanoparticles to base fluids leads to three different results: augmentations of the heat transfer coefficient and critical heat flux, a decrease or no change. Mukherjee et al. found that 55% of the total searched papers agreed with the augmentation, 37% said deterioration, and the remaining 8% reported inconclusive or no change. Al2O3 is the most used type of nanoparticle with 35% of the studies, followed by CuO, TiO2, CNT, SiO2, ZnO, ZrO2, and Fe3O4. Water is mostly used as a base fluid, with 84% of the studies, followed by an ethylene glycol–water mixture, refrigerant and others. For this reason, the study presented by the authors has relevance considering the mixtures suggested.

However, there needs to be some more information regarding the TiO2 used. Rutile and anatase have some differences between the 2 types: their crystalline structure; and rutile has a greater dispersion capacity, hardness and density. Also, the geometry (sphere or rod) and the purity are important to reach some conclusions. Currently it is already evident that these properties affect the stability and thermal properties of nanofluids.

The authors do not specify if physical properties such as viscosity, density, boiling point and surface tension were measured in the nanofluids studied. Did any of these properties have a significant change between different concentrations of nanomaterials and surfactants? It would be important to have some comment on this.

Some corrections must be made to the heat flux units in the text (kw/m2 to kW/m2).

1. Mukherjee, S.; Ebrahim, S.; Mishra, P.C.; Ali, N.; Chaudhuri, P. A Review on Pool and Flow Boiling Enhancement Using Nanofluids: Nuclear Reactor Application. Processes 2022, 10, 177

Author Response

Dear reviewer:

Thank you for your decision and constructive comments on my manuscript. We have carefully considered the suggestion of the Reviewer and made some changes. We have tried our best to improve and made some changes to the manuscript.

The yellow part has been revised according to your comments. Revision notes, point-to-point, are given as follows:

1. We agree with the reviewer that measuring more specific physical properties of nanofluids would be helpful. However, we believe that physical properties such as viscosity, density, boiling point, and surface tension all significantly influence on heat transfer performance of fluids. We do not make detail explore this section for two reasons:

1) Measuring every physical property needs a lot of money and time. We have not finished this part right now. However, we can narrow the properties we need in further study with experimental results in this research. We believe it will do the same to help other researchers.

2)The main point of this paper is to experimentally investigate how regular heat transfer enhancement ways work on non-zeotropic mixtures. And nanofluids were a part of our enhancing means. What is more, the results showed a deterioration in heat transfer performance. So most part of the properties explanation is about basic fluids instead of nanofluids.

We appreciate the reviewer’s insightful suggestion and apologize that we do not rewrite this section.

However, we mentioned this question in the further study in the last part. And we add the bubble behavior of different concentration non-azeotropic mixtures in Figure 6 to show the change in properties.

2. We appreciate your opinion on the writing errors. We have checked on the whole essay to make sure there were no errors as possible.

We would like to express our great appreciation to you for your comments on our paper. Looking forward to hearing from you soon.

Thank you and best regards!

Yours sincerely,

Chen Xu
